# 3D Surface Topographic Optical Scans Yield Highly Reliable Global Spine Range of Motion Measurements in Scoliotic and Non-Scoliotic Adolescents

**DOI:** 10.3390/children9111756

**Published:** 2022-11-16

**Authors:** Kira Page, Caroline Gmelich, Ankush Thakur, Jessica H. Heyer, Howard J. Hillstrom, Benjamin Groisser, Kyle W. Morse, Don Li, Matthew E. Cunningham, M. Timothy Hresko, Roger F. Widmann

**Affiliations:** 1Department of Pediatric Orthopaedics, Hospital for Special Surgery, New York, NY 10021, USA; 2Hospital for Special Surgery Research Institute, Hospital for Special Surgery, New York, NY 10021, USA; 3Faculty of Mechanical Engineering, Technion—Israel Institute of Technology, Haifa 320003, Israel; 4Department of Spine Surgery, Hospital for Special Surgery, New York, NY 10021, USA; 5Department of Pediatric Orthopaedics, Boston Children’s Hospital, Boston, MA 02115, USA

**Keywords:** adolescent idiopathic scoliosis, spine range of motion, scoliosis screening, scoliosis

## Abstract

Background: Adolescent idiopathic scoliosis results in three dimensional changes to a patient’s body, which may change a patient’s range of motion. Surface topography is an emerging technology to evaluate three dimensional parameters in patients with scoliosis. The goal of this paper is to introduce novel and reliable surface topographic measurements for the assessment of global coronal and sagittal range of motion of the spine in adolescents, and to determine if these measurements can distinguish between adolescents with lumbar scoliosis and those without scoliosis. Methods: This study is a retrospective cohort study of a prospectively collected registry. Using a surface topographic scanner, a finger to floor and lateral bending scans were performed on each subject. Inter- and intra-rater reliabilities were assessed for each measurement. ANOVA analysis was used to test comparative hypotheses. Results: Inter-rater reliability for lateral bending fingertip asymmetry (LBFA) and lateral bending acromia asymmetry (LBAA) displayed poor reliability, while the coronal angle asymmetry (CAA), coronal angle range of motion (CAR), forward bending finger to floor (FBFF), forward bending acromia to floor (FBAF), sagittal angle (SA), and sagittal angle normalized (SAN) demonstrated good to excellent reliability. There was a significant difference between controls and lumbar scoliosis patients for LBFA, LBAA, CAA and FBAF (*p*-values < 0.01). Conclusion: Surface topography yields a reliable and rapid process for measuring global spine range of motion in the coronal and sagittal planes. Using these tools, there was a significant difference in measurements between patients with lumbar scoliosis and controls. In the future, we hope to be able to assess and predict perioperative spinal mobility changes.

## 1. Introduction

Adolescent idiopathic scoliosis (AIS) is a common pediatric spinal deformity evaluated by pediatricians and surgeons [1]. The spine is the primary axis for truncal rotation, which enables both lateral and forward bending movements [2]. Daily activities require complex functioning of the spine in three dimensions [3]. Because AIS is a deformity that affects all three planes, it may restrict or alter spine movement and/or pelvic alignment [4,5]. Considering the impact of spine range of motion (ROM) on daily activities, sports, and overall health, it is important that clinicians have reliable measurements of spine motion.

Historically, motion of the spine has been measured using a variety of different techniques, including fingertip-to-floor measurements, the Schober and modified Schober tests, triflexometer, dual inclinometers, electrogoniometer, and radiography [3,6,7,8,9]. These techniques vary in inter- and intra-rater reliability, and several are user-dependent [3,6,7,10,11,12,13,14]. While dynamic radiographs have been used as a gold standard with which to compare surface ROM measurements, dynamic radiographs require additional exposure to ionizing radiation, and reliability of dynamic radiographs not well-studied since this requires repeated exposure to ionizing radiation [4,6,9].

The 3dMD system is a three-dimensional topographical scanner that has established accuracy and reliability in multiple fields of clinical medicine [15,16,17]; other topographical scanners have demonstrated high reliability in the assessment of scoliosis patients [18]. Surface topographic measurements can be used to evaluate global spine ROM if accuracy and reliability of these parameters can be established. The goal of this paper is to introduce novel and reliable surface topographic model-derived measurements for the assessment of global spine coronal and sagittal ROM in adolescents. Secondary aims of this study include discerning if these measurements can distinguish between adolescents with and without lumbar curves. We hypothesize that there will be a difference in the measurements between patients with and without lumbar curves.

## 2. Materials and Methods

### 2.1. Subjects

Subject data was obtained from the Spinal Alignment Registry (SAR); all subjects in the SAR were recruited from the Pediatric Orthopaedic Department at the Hospital for Special Surgery (HSS). The SAR was approved by the institutional review board at HSS. Informed consent and assent were obtained from subjects or subjects’ parents (for all participants under 18 years of age). Subjects deemed “patients,” were those scheduled for an assessment of idiopathic scoliosis with an orthopaedic surgeon and were between 11–21 years old at the time of assessment and underwent EOS biplanar scoliosis radiography confirming scoliosis. Subjects deemed “controls” were those recruited from the pediatric Sports Medicine Department who had no spinal deformity detected on exam. All patients and controls had a standard clinical assessment and then underwent a 3dMD scan, as described below. For the reliability evaluation, a subset of 46 SAR subjects (controls and patients) were evaluated using repeated measurements. For the evaluation of controls versus patients, patients were included if they had AIS and a lumbar curve with a Cobb angle greater than 25 degrees. This included patients with thoracic curves of all Lenke sub-types, as long as there was a lumbar curve of greater than 25 degrees.

### 2.2. Surface Topographic Scanner

The 3dMD topographic scanner (3dMD, Atlanta, GA, USA) is a high resolution optical stereophotogrammetric system comprising 30 machine-vision cameras mounted within an 8′ by 10′ enclosure. The 3dMD topographic scanner produces whole body high-precision 3D dense surface images with a linear accuracy range of 0.7 mm or better. The system is able to capture 3D video at 10 frames per second, with each scan taking 1.7 milliseconds, enabling the capture of dynamic movements to measure range of motion. The rapid capture time minimizes motion artifact within the range of normal human movements. The available scanning volume is able to scan all ranges of standing and bending postures.

### 2.3. Scan Protocol

Subjects changed into form-fitting clothing: low-waisted shorts and hair nets for all participants, and a custom, open-back bra for females. Subjects stood in the center of the 3dMD optical scanner in a relaxed stance.

For the finger to floor scan, the subject is first positioned on a block in the Adams forward bending posture (Figure 1A), and instructed to reach to the floor to their maximum extent (Figure 1B), at which point the scan is performed.

For the lateral bending scans, the subject is positioned in the A-pose (feet in a wide base of support, with their arms raised approximately 45 degrees laterally away from the subject’s body, Figure 2A). Once in this position, the subject is instructed to bend maximally to the left (Figure 2B) and then to the right (Figure 2C) with their fingers reaching to the floor without any twisting of shoulders or pelvis. The frames of maximum bending are identified to be reconstructed and processed.

### 2.4. Automated Analysis

Surface scan images of relevant frames are reconstructed into 3D full body models by 3dDM software. Surface scans were processed using a fully automated analysis pipeline previously described by Groisser et al. to enable objective surface landmark detection and measurements [19]. Briefly, an annotated template mesh model is fitted to each surface scan, allowing the same landmarks to be identified on all patient scans. Landmarks of interests including PSIS, ASIS, C7, fingertips, and acromioclavicular joints are indicated on this template model with can then be mapped onto each surface scan automatically. Distance and angular measurements are computed from 3D coordinates of each landmark. This process enables scans to be processed and measurements to be extracted without human error and manual analysis.

### 2.5. Reliability

All study subjects were scanned three times each by two raters in a randomized order. The first scan was obtained as per protocol above. The subject was then re-scanned without changing foot position (test–retest). The subject was then instructed to leave the scanning area and return to re-scan (remove-replace). These three scans were then repeated by a second investigator.

### 2.6. Measurements

The authors propose measurements of global spine ROM based on automated identification of standard anatomic landmarks including posterior superior iliac spine (PSIS), C7, acromioclavicular (AC) joints, and fingertips. Measurement of ROM and asymmetry were performed using linear distance measurements and angular measurements. Bending distance and angle measurements are computed on each scan (A pose, left bending, right bending, and forward bending) independently using the automated analysis pipeline described above, and then range of motion and asymmetry measurements are computed using the equations described in Table 1. Lateral bending measurements were made using the lateral bend scans (Figure 3), while forward bending measurements were made using the forward bend scan (Figure 4). The A pose was used as a reference for neutral position for all relevant measurements.

### 2.7. Statistical Analysis

Intraclass correlation coefficients (ICC) and their 95% confidence intervals (CI) were calculated for intra-rater (test–retest, remove-replace) and inter-rater conditions to assess measurement reliability. Intra-rater results are reported as the mean of the two raters. ICC values < 0.50, 0.50–0.74, 0.75–0.90, >0.90 were used to indicate poor, moderate, good, or excellent reliability, respectively [20]. Comparison testing for patients and controls was performed using analysis of variance (ANOVA). Analyses were performed using SPSS software (International Business Machines, Armonk, NY, USA, version 22). Results were considered significant at *p* < 0.05.

## 3. Results

### 3.1. Reliability

The cohort to establish reliability consisted of 46 subjects. There were 20 controls and 26 patients. The average age was 14.6 years (standard deviation [SD] 2.4) for controls and 14.5 years (SD 2.7) for patients. Controls had a mean body mass index (BMI) of 21.7 kg/m^2^ (SD 3.8) and patients had a mean BMI of 20.6 kg/m^2^ (SD 4.3) (Table 2).

Results of the reliability evaluation are reported in Table 3. Test–retest reliability was higher than remove-replace for each measurement. In regard to inter-rater reliability, the lateral bending fingertip asymmetry (LBFA) and lateral bending acromia asymmetry (LBAA) demonstrated poor reliability, while the coronal angle asymmetry (CAA), coronal angle ROM (CAR), forward bending finger to floor (FBFF), forward bending acromia to floor (FBAF), sagittal angle (SA), sagittal angle normalized (SAN) demonstrated good to excellent reliability.

### 3.2. Scoliosis Patients Versus Controls

The comparison analysis was completed on a cohort of 58 patients and 37 controls. The average age was 14.6 years (SD 2.2) and average body mass index (BMI) of 21.1 kg/m^2^ (SD 4.6). The average thoracic Cobb angle was 49.5 degrees (SD 17.1), and average lumbar Cobb angle was 42.1 degrees (SD 13.4). The control group had an average age of 14.3 years (SD 2.36) and average BMI of 21.9 kg/m^2^ (SD 3.9) (Table 4).

In the lateral bend scan, there was a significant difference between patients and controls for LBFA, LBAA, and CAA (*p*-values 0.011, 0.001, and <0.001, respectively). In the forward bend, only FBAF distance showed a significant difference between patients and controls (*p* = 0.018) (Table 5).

## 4. Discussion

This study demonstrates that 3D surface topographic scanning technology paired with sophisticated software analysis provides a highly reliable process for measuring global spine ROM in the coronal and sagittal planes in adolescents. Using these measurements, we were able to differentiate between patients with and without scoliosis.

Current methods of evaluating sagittal plane motion include finger to floor measurements, the Schober and modified Schober tests, inclinometers, and electrogoniometers. The sagittal plane ROM measurements in our study demonstrated good to excellent inter-rater reliability with both linear measurements (FBFF) and angular measurements (SA and SAN). Our methodology uses automatic landmarking, eliminating the need for manual identification of landmarks. Prior reliability studies have shown high degrees of variation for the finger to floor test, which is likely limited by its failure to account for scapulothoracic and shoulder motion [6,10,11,12,13,14]. This limitation holds true for both the manual and surface topographic means of measuring. However, advantages of our methodology are that we can measure the distance from finger to floor in addition to acromion to floor quickly, and we normalize to the patient’s height. The Schober and modified Schober tests are limited in that they only evaluate lumbar flexion, not global spine motion. Reports on reliability of these tests vary from poor to excellent, which may be attributed to the large amount of error introduced by misplacement of the markers in the exam [7,10,11,14,21]. Additionally, the Schober test has been shown to have weak correlation to radiographic mobility [21]. Dual inclinometers also only evaluate lumbar flexion, but they eliminate the effect of pelvic flexion, which makes it more attractive as an assessment of lumbar motion [6,11]. However, reliability of the dual inclinometers is heavily affected by training [22]. Lastly, electrogoniometers have been shown to have excellent inter-rater reliability to assess full spine motion at 0.96 [3]. Hresko et al., demonstrated that the Schober test, inclinometers, and electrogoniometers did not correlate with one another’s measurements, indicating that their values are not necessarily correlating to a patient’s spinal motion [9]. Given our study’s excellent reliability values for the FBFF, SA and SAN measurements, we suggest use of these metrics instead of the aforementioned tests.

Measuring lateral bending, our study found good reliability for both angular measurements CAA and CAR. The previously described ways to assess lateral bending includes side bending finger to floor measurements and the Moll lateral flexion test [11]. In the literature, lateral bending finger to floor has poor to good reliability, with intra-rater reliability ranging from 0.64–0.88 and inter-rater reliability ranging from 0.47–0.87 [10,14]. The Moll lateral flexion test is similar to the Schober test, modified for the lateral bend; the inter-rater variability is 10–12%, while the intra-rater variability is 8–10%. While our lateral bending measurements are not as reliable as our forward flexion measurements, this is probably due to variation in patient posture in side bending secondary to the complex three-dimensional relationships between vertebral bodies that account for global coronal spine ROM. The reliability of measurements is likely highly dependent upon patient cooperation and effort. This study presents these two angular measurements as reliable ways to assess lateral bending that does not rely on finding patients’ bony landmarks as is required for the Moll test.

Based upon our study, angle-based measurements were more reliable than distance-based linear measurements in the assessment of global spine ROM. With regard to coronal plane measurements, the two angular measurements were found to be more reliable than the linear measurements. In the sagittal plane, the angular measurements were superior to the FBAF measurement, and comparable to the FBFF. Angular measurements eliminate the confounding component of height or extremity length that may vary between patients and require normalization for clinical use. This aligns with a study by Eyvazov et al., which evaluated patients with Lenke 5 curves and found that only the lateral bending angle was reflective of curve severity, but not the linear measurements tested [4].

This study demonstrates that our global coronal and sagittal plane spine ROM measurements derived from surface scans were able to differentiate between patients with and without lumbar scoliosis in our database. Lateral bending measurements of LBFA, LBAA, CAA, and the forward bending measurement of FBAF were significantly different between subjects with lumbar scoliosis and those without. In our study, the forward bend acromia to floor measurement demonstrated significant differences between the scoliosis group and controls, however other sagittal plane measurements did not reveal significant differences. We hypothesize that this may be due to change in shoulder elevation between the cohorts with and without scoliosis, which may not have been detected in the forward bend fingertip to floor measurement. Prior studies have been performed looking at the responsiveness of lumbar motion tests to low back pain patients, and have found that the finger to floor test is responsive to low back pain, but not the Schober test [6,23]. Eyvazov et al., did not find that lateral bending or forward bending finger to floor correlated to lumbar curve severity in Lenke 5 patients, but that lateral bending angle did decrease in patients with more severe curves [4]. However, Hresko et al., who did not limit their evaluation to Lenke 5 scoliosis patients, did not find that lateral bend angle correlated to curve severity. Furthermore, none of the forward bend tests including the Schober test, dual inclinometers or electrogoniometer methods were correlated with curve severity [9].

This study has several limitations. One limitation is that this study is specific to the use of 3dMD software and our measurement algorithms; however, our measurement algorithms can be applied to any surface topographic capture system. We chose to limit our patient cohort to those with lumbar curves >25 degrees, since we believed this would have the greatest impact on side bending. Although we did require a minimum lumbar threshold, the average thoracic curve was >40 degrees, indicating that there were both thoracic and lumbar curves in these patients. Future studies can compare the impact of structural thoracic versus structural lumbar curves on patient’s motion. Clinical studies with longer term follow-up on global spine ROM are in process.

In future studies, we plan to examine the relationship between radiographic side bending measurements and surface topographic side bending measurements. We hope to decrease the use of ionizing radiation in the assessment of mobility in AIS patients through such analysis. Additionally, we plan to further evaluate the differences in controls versus patients with scoliosis, and discern if there are tests that may be useful in screening for scoliosis at different thresholds of curvature. With continuing data collection, we hope to be able to assess and quantify perioperative spinal mobility changes and develop a predictive tool for such changes.

## 5. Conclusions

Surface topographic scanning paired with custom software, is a fast and reliable method for the evaluation of global spine ROM in the sagittal and coronal planes, and does not rely on fiducial landmarks, identification of bony landmarks, radiation, or use of user-dependent equipment.

Angle based measurements are more reliable than linear measurements when using surface topographical analysis of global spine ROM.

Coronal and sagittal measurements of global spine ROM demonstrate significant differences between a cohort of healthy controls and subjects with AIS (greater than 25° lumbar Cobb angles).

## Figures and Tables

**Figure 1 children-09-01756-f001:**
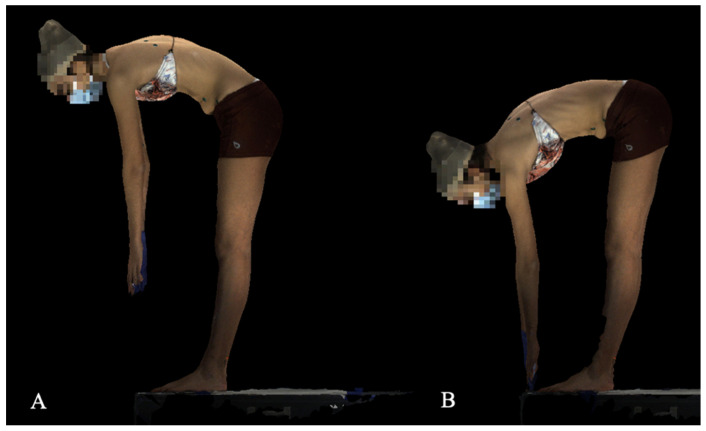
Finger to floor position (**A**). Subject in the Adams forward bend position. (**B**). The finger to floor scan, where the subject is reaching to the floor to his or her maximum extent. The patient is positioned on a block in the event they can reach beyond their toes.

**Figure 2 children-09-01756-f002:**
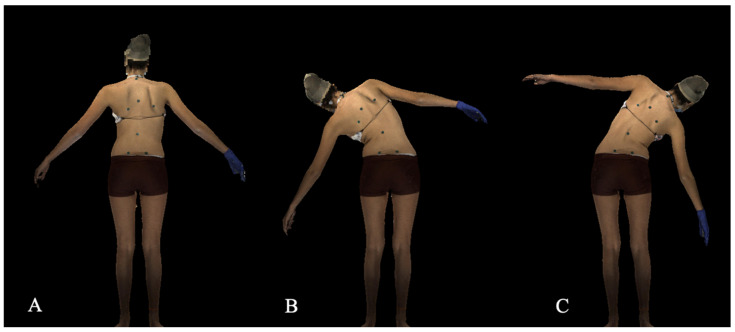
Lateral bend position (**A**). The A pose, with arms raised approximately 45 degrees laterally away from the body. (**B**). Subject posing with maximal bend to the left. (**C**). Subject posing with maximal bend to the right.

**Figure 3 children-09-01756-f003:**
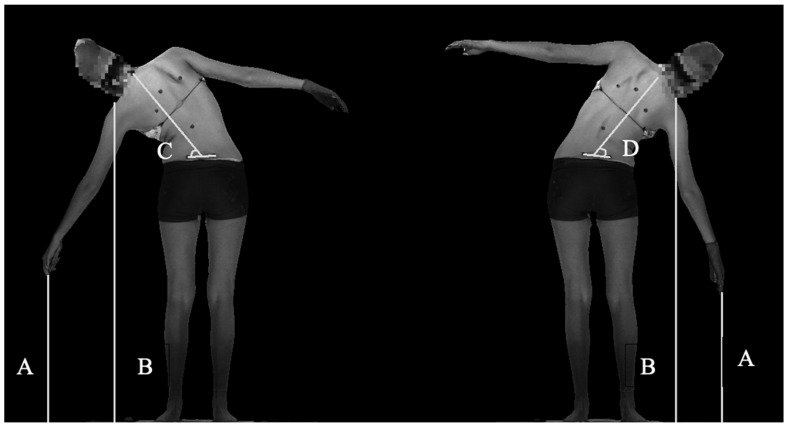
Lateral bending measurements Line A depicts distance from fingertips to floor; line B depicts distance from acromioclavicular joint to floor. Angles C and D represent the left bend coronal angle and right bend coronal angle, respectively.

**Figure 4 children-09-01756-f004:**
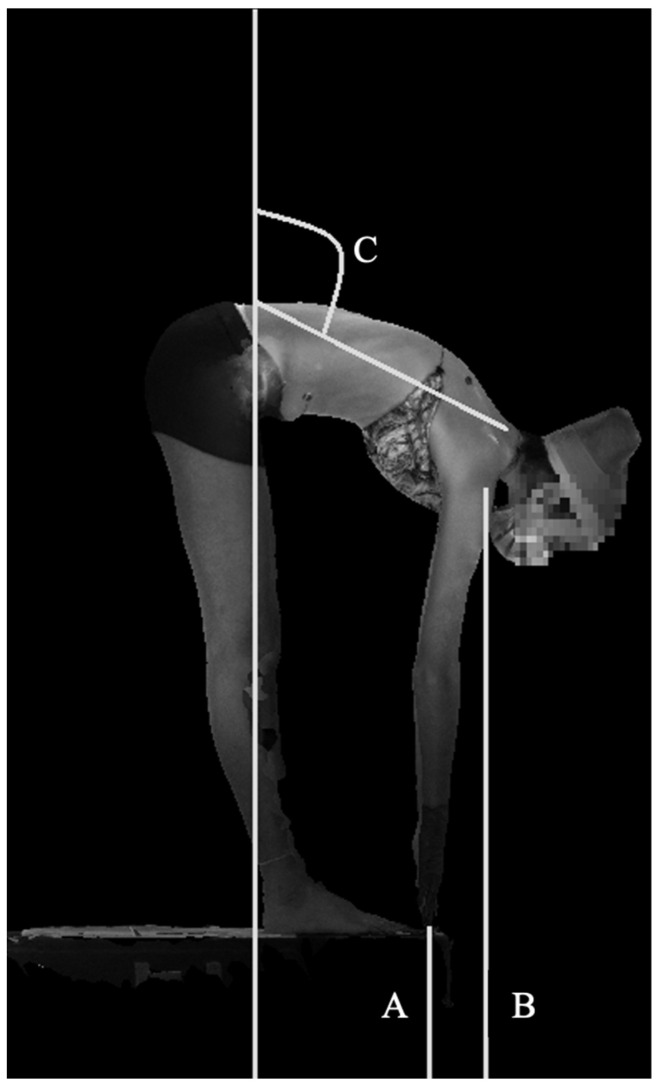
Forward bending measurements Line A depicts distance from fingertips to floor; line B depicts distance from acromioclavicular joint to floor. Angle C represents the sagittal angle.

**Table 1 children-09-01756-t001:** Lateral and forward bending measurements.

**Lateral Bending Measurements**	**Description**	**Calculation**
Lateral Bend Fingertip Asymmetry	Asymmetry of left vs. right bend fingertips to floor	200 × |(finger tips right − finger tips left)|/(finger tips right + finger tips left)
Lateral Bend Acromia Asymmetry	Asymmetry of left vs. right bend acromia to floor	200 × |(Right AC height right − left AC height left)|/(Right AC height right + left AC height left)
Coronal Angle Asymmetry	Asymmetry of left vs. right coronal angle (angle between C7 to PSIS midpoint and line between PSIS)	200 × |(coronal angle right − coronal angle left)|/(coronal angle right + coronal angle left)
Coronal Angle ROM	Left + right coronal angle (summation of angles from C7 to PSIS midpoint and line between PSIS in left and right bends)	(coronal angle right + coronal angle left)
**Forward Bending Measurements**	**Description**	**Calculation**
Forward Bend Finger to Floor	Finger to floor distance in maximum forward flexion. Normalized to patient height.	(finger to floor distance)/(patient height)
Forward Bend Acromia to Floor	Average of acromioclavicular joint distance to floor in maximum forward flexion. Normalized to height.	(average distance from right AC joint to floor and left AC joint to floor)/(height)
Sagittal Angle	Maximum forward bend angle measured from C7 to PSIS midpoint, referenced to the line perpendicular to the floor.	(maximum forward bend angle)
Sagittal Angle Normalized	Maximum forward bend angle measured from C7 to PSIS midpoint, subtracting reference angle in A-pose.	(Sagittal angle forward − sagittal angle A pose)

ROM: Range of motion; PSIS: posterior superior iliac spine; AC: acromioclavicular; C7: 7th cervical vertebrae.

**Table 2 children-09-01756-t002:** Demographics, Reliability Cohort.

Reliability	Controls, n = 20	Patients, n = 26
Sex		
Males, n (%)	11 (55)	12 (46.2)
Age, mean (range, SD), years	14.6 (11–20, 2.4)	14.5 (11–21, 2.7)
BMI, mean (range, SD), kg/m^2^	21.7 (16.8–28.7, 3.8)	20.6 (15.9–35.9, 4.3)

BMI: Body mass index, SD: standard deviation.

**Table 3 children-09-01756-t003:** Inter-rater and intra-rater reliability of lateral and forward bending measurements.

Measurements	Intra-Rater Reliability, ICC (95%CI)		Inter-Rater Reliability, ICC (95%CI)
	Test–Retest	Remove-Replace	
Lateral Bending			
LBFA	0.448, 0.481 (0.165–0.677)	0.233, 0.457 (−0.069–0.682)	0.496 (0.236–0.689)
LBAA	0.509, 0.564 (0.254–0.736)	0.176, 0.588 (−0.128–0.752)	0.433 (0.155–0.649)
CAA	0.576, 0.769 (0.339–0.867)	0.601, 0.738 (0.372–0.848)	0.756 (0.594–0.859)
CAR	0.910, 0.965 (0.842–0.981)	0.795, 0.819 (0.655–0.897)	0.783 (0.636–0.876)
Forward Bending			
FBFF	0.984, 0.990 (0.971–0.995)	0.985. 0.985 (0.968–0.993)	0.984 (0.970–0.991)
FBAF	0.939, 0.958 (0.891–0.977)	0.799, 0.836 (0.660–0.907)	0.746 (0.580–0.852)
SA	0.666, 0.750 (0.458–0.859)	0.413, 0.995 (0.132–0.995)	0.994 (0.988–0.996)
SAN	0.973, 0.973 (0.950–0.986)	0.957, 0.981 (0.921–0.990)	0.977 (0.957–0.987)

LBFA: lateral bending fingertip asymmetry, LBAA: lateral bending acromia asymmetry, CAA: coronal angle asymmetry, CAR: coronal angle range of motion, FBFF: forward bending finger to floor, FBAF: forward bending acromia to floor, SA: sagittal angle, SAN: sagittal angle normalized, ICC: Intraclass correlation coefficient; CI: confidence interval. White shaded boxes indicate excellent ICCs, light grey indicates good to moderate ICCs, and dark grey indicates poor ICCs.

**Table 4 children-09-01756-t004:** Demographics, Scoliosis patients vs. control cohort.

Scoliosis Patients vs. Controls	Controls, n = 37	Patients, n = 58
Sex		
Males, n (%)	23 (62.2)	21 (36.2)
Age, mean (range, SD), years	14.3 (11–20, 2.4)	14.6 (11–21, 2.2)
BMI, mean (range, SD), kg/m^2^	21.9 (16.8–29.7, 3.9)	21.1 (15.3–35.9, 4.6)
Thoracic Cobb angle, average (range, SD), degrees	N/A	49.5 (15.2–83.1, 17.1)
Lumbar Cobb angle, average (range, SD), degrees	N/A	42.1 (25.1–86.3, 13.4)

BMI: Body mass index, SD: standard deviation. N/A: Not applicable.

**Table 5 children-09-01756-t005:** Comparison of AIS patients to controls.

	Patients, n = 58		Controls, n = 37		
	Mean	SD	Mean	SD	*p*-Value
**Lateral Bending Measurements**					
LBFA	10.80	9.01	6.35	6.40	**0.011**
LBAA	3.56	2.81	1.83	1.74	**0.001**
CAA	22.73	17.40	9.34	6.62	**≤0.001**
CAR (degrees)	68.91	18.99	73.68	12.15	0.18
**Forward Bending Measurements ***					
FBFF	0.076	0.062	0.07	0.054	0.63
FBAF	0.45	0.07	0.41	0.07	**0.018**
SA (degrees)	65.56	13.36	60.07	16.22	0.086
SAN (degrees)	59.14	13.13	54.19	16.27	0.12

LBFA: lateral bending fingertip asymmetry, LBAA: lateral bending acromia asymmetry, CAA: coronal angle asymmetry, CAR: coronal angle range of motion, FBFF: forward bending finger to floor, FBAF: forward bending acromia to floor, SA: sagittal angle, SAN: sagittal angle normalized, SD: standard deviation. * Patient n = 56, control n = 37. Bolded values indicate significance at *p* < 0.05.

## Data Availability

The data collected in this study is stored on the secure drive at Hospital for Special Surgery and can be made available on request.

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
