# Peer review of "3D Surface Topographic Optical Scans Yield Highly Reliable Global Spine Range of Motion Measurements in Scoliotic and Non-Scoliotic Adolescents"

_children, 2022, doi:10.3390/children9111756_

Round 1

Reviewer 1 Report

The paper needs some clarifications:

I feel that the authors could rephrase the title including “scoliosis” somewhere. 

3D Surface Topographic Optical Scans Yield Highly Reliable Global Spine Range of Motion Measurements “in healthy and scoliotic” Adolescents 

1 – in M&M could you clarify why you recommend raising the arms around 45° and not leaving them along the body. This is not very a “natural” position. 

2 – For the “evaluation of controls versus patients, patients were included if they had AIS and a lumbar curve with a Cobb angle greater than 25 degrees.” Could you clarify the curves types of your patients : thoracic + lumbar (1C or 3C) and pure lumbar (5C) ? What I understand is that you wanted to compare patients with AIS With a lumbar curve (with or without a thoracic curve) versus healthy subjects. It is unclear in M&M. 

Reviewer 2 Report

Although this study suggests a promising and moreover interesting approach, there are some flaws and inconsistencies  

Page 1 line 29 taking the setup of the study into account it is very optimistic/ambitious to say that you were able to distinguish between scoliosis and non scoliosis 

Page2 line 64 why is this blacked out in the manuscript – not acceptable, same applies to line 65

Table2 is confusing and needs to be restructured

It remains unclear how the measurements were performed? Software based? The whole process has to be described adequately

Check for typos

Round 2

Reviewer 2 Report

ad response #1: there were significant differences - according to the setup you can't just state, that you are able to distinguish between scoliotic and non-scoliotic patients.

ad response #2: given that the affiliations were openly documented in the manuscript, this is not much of a surprise

ad response #3: sufficient
